# Unsupervised Learning of Object Structure and Dynamics from Videos

**Matthias Minderer**[*]     **Chen Sun     Ruben Villegas     Forrester Cole**
**Kevin Murphy     Honglak Lee**
Google Research
`{mjlm, chensun, rubville, fcole, kpmurphy, honglak}@google.com`

## Abstract

Extracting and predicting object structure and dynamics from videos without supervision is a major challenge in machine learning. To address this challenge, we adopt a keypoint-based image representation and learn a stochastic dynamics model of the keypoints. Future frames are reconstructed from the keypoints and a reference frame. By modeling dynamics in the keypoint coordinate space, we achieve stable learning and avoid compounding of errors in pixel space. Our method improves upon unstructured representations both for pixel-level video prediction and for downstream tasks requiring object-level understanding of motion dynamics. We evaluate our model on diverse datasets: a multi-agent sports dataset, the Human3.6M dataset, and datasets based on continuous control tasks from the DeepMind Control Suite. The spatially structured representation outperforms unstructured representations on a range of motion-related tasks such as object tracking, action recognition and reward prediction.

## 1   Introduction

Videos provide rich visual information to understand the dynamics of the world. However, extracting a useful representation from videos (e.g. detection and tracking of objects) remains challenging and typically requires expensive human annotations. In this work, we focus on unsupervised learning of object structure and dynamics from videos.

One approach for unsupervised video understanding is to learn to predict future frames [17, 16, 9, 15, 24, 30, 8, 3, 14]. Based on this body of work, we identify two main challenges: First, it is hard to make pixel-level predictions because motion in videos becomes highly stochastic for horizons beyond about a second. Since semantically insignificant deviations can lead to large error in pixel space, it is often difficult to distinguish good from bad predictions based on pixel losses. Second, even if good pixel-level prediction is achieved, this is rarely the desired final task. The representations of a model trained for pixel-level reconstruction are not guaranteed to be useful for downstream tasks such as tracking, motion prediction and control.

Here, we address both of these challenges by using an explicit, interpretable keypoint-based representation of object structure as the core of our model. Keypoints are a natural representation of dynamic objects, commonly used for face and pose tracking. Training keypoint detectors, however, generally requires supervision. We learn the keypoint-based representation directly from video, without any supervision beyond the pixel data, in two steps: first encode individual frames to keypoints, then model the dynamics of those points. As a result, the representation of the dynamics model is spatially structured, though the model is trained only with a pixel reconstruction loss. We show that enforcing spatial structure significantly improves video prediction quality and performance for tasks such as action recognition and reward prediction.

---

[*]Google AI Resident

By decoupling pixel generation from dynamics prediction, we avoid compounding errors in pixel space because we never condition on predicted pixels. This approach has been shown to be beneficial for supervised video prediction [25]. Furthermore, modeling dynamics in keypoint coordinate space allows us to sample and evaluate predictions efficiently. Errors in coordinate space are more meaningful than in pixel space, since distance between keypoints is more closely related to semantically relevant differences than pixel-space distance. We exploit this by using a best-of-many-samples objective [4] during training to achieve stochastic predictions that are both highly diverse and of high quality, outperforming the predictions of models lacking spatial structure.

Finally, because we build spatial structure into our model *a priori*, its internal representation is biased to contain object-level information that is useful for downstream applications. This bias leads to better results on tasks such as trajectory prediction, action recognition and reward prediction.

Our contributions are: (1) a novel architecture and optimization techniques for unsupervised video prediction with a structured internal representation; (2) a model that outperforms recent work [8, 28] and our unstructured baseline in pixel-level video prediction; (3) improved performance vs. unstructured models on downstream tasks requiring object-level understanding.

## 2 Related work

**Unsupervised learning of keypoints.** Previous work explores learning to find keypoints in an image by applying an autoencoding architecture with keypoint-coordinates as a representational bottleneck [12, 33]. The bottleneck forces the image to be encoded in a small number of points. We build on these methods by extending them to the video setting.

**Stochastic sequence prediction.** Successful video prediction requires modeling uncertainty. We adopt the VRNN [6] architecture, which adds latent random variables to the standard RNN architecture, to sample from possible futures. More sophisticated approaches to stochastic prediction of keypoints have been recently explored [31, 21], but we find the basic VRNN architecture sufficient for our applications.

**Unsupervised video prediction.** A large body of work explores learning to predict video frames using only a pixel-reconstruction loss [18, 20, 17, 9, 24, 7]. Most similar to our work are approaches that perform deterministic image generation from a latent sample produced by stochastic sampling from a prior conditioned on previous timesteps [8, 3, 14]. Our approach replaces the unstructured image representation with a structured set of keypoints, improving performance on video prediction and downstream tasks compared with SVG [8] (Section 5).

Recent methods also apply adversarial training to improve prediction quality and diversity of samples [22, 14]. EPVA [28] predicts dynamics in a high-level feature space and applies an adversarial loss to the predicted features. We compare against EPVA and show improvement without adversarial training, but adversarial training is compatible with our method and is a promising future direction.

**Video prediction with spatially structured representations.** Like our approach, several recent methods explore explicit, spatially structured representations for video prediction. Xu et al. [29] proposed to discover object parts and structure by watching how they move in videos. Vid2Vid [27] proposed a video-to-video translation network from segmentation masks, edge masks and human pose. The method is also used for predicting a few frames into the future by predicting the structure representations first. Villegas et al. [25] proposed to train a human pose predictor and then use the predicted pose to generate future frames of human motion. In [26], a method is proposed where future human pose is predicted using a stochastic network and the pose is then used to generate future frames. Recent methods on video generation have used spatially structured representations for video motion transfer between humans [1, 5]. In contrast, our model is able to find spatially structured representation without supervision while using video frames as the only learning signal.

## 3 Architecture

Our model is composed of two parts: a keypoint detector that encodes each frame into a low-dimensional, keypoint-based representation, and a dynamics model that predicts dynamics in the keypoint space (Figure 1).

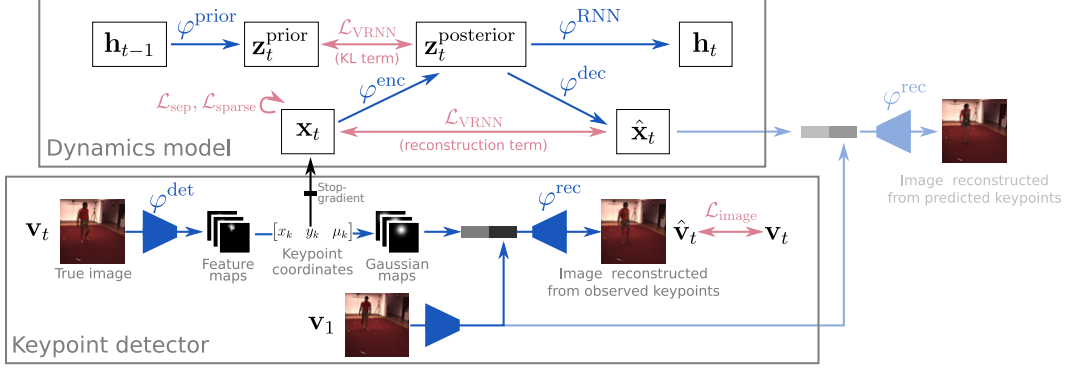

Figure 1: Architecture of our model. Variables are black, functions blue, losses red. Some arrows are omitted for clarity, see Equations 1 to 4 for details.

## 3.1 Unsupervised keypoint detector

The keypoint detection architecture is inspired by [12], which we adapt for the video setting. Let $\mathbf{v}_{1:T} \in \mathbb{R}^{H \times W \times C}$ be a video sequence of length $T$. Our goal is to learn a keypoint detector $\varphi^{\text{det}}(\mathbf{v}_t) = \mathbf{x}_t$ that captures the spatial structure of the objects in each frame in a set of keypoints $\mathbf{x}_t$.

The detector $\varphi^{\text{det}}$ is a convolutional neural network that produces $K$ feature maps, one for each keypoint. Each feature map is normalized and condensed into a single $(x, y)$-coordinate by computing the spatial expectation of the map. The number of heatmaps $K$ is a hyperparameter that represents the maximum expected number of keypoints necessary to model the data.

For image reconstruction, we learn a generator $\varphi^{\text{rec}}$ that reconstructs frame $\mathbf{v}_t$ from its keypoint representation. The generator also receives the first frame of the sequence $\mathbf{v}_1$ to capture the static appearance of the scene: $\mathbf{v}_t = \varphi^{\text{rec}}(\mathbf{v}_1, \mathbf{x}_t)$. Together, the keypoint detector $\varphi^{\text{det}}$ and generator $\varphi^{\text{rec}}$ form an autoencoder architecture with a representational bottleneck that forces the structure of each frame to be encoded in a keypoint representation [12].

The generator is also a convolutional neural network. To supply the keypoints to the network, each point is converted into a heatmap with a Gaussian-shaped blob at the keypoint location. The $K$ heatmaps are concatenated with feature maps from the first frame $\mathbf{v}_1$. We also concatenate the keypoint-heatmaps for the first frame $\mathbf{v}_1$ to the decoder input for subsequent frames $\mathbf{v}_t$, to help the decoder to "inpaint" background regions that were occluded in the first frame. The resulting tensor forms the input to the generator. We add skip connections from the first frame of the sequence to the generator output such that the actual task of the generator is to predict $\mathbf{v}_t - \mathbf{v}_1$.

We use the mean intensity $\mu_k$ of each keypoint feature map returned by the detector as a continuous-valued indicator of the presence of the modeled object. When converting keypoints back into heatmaps, each map is scaled by the corresponding $\mu_k$. The model can use $\mu_k$ to encode the presence or absence of individual objects on a frame-by-frame basis.

## 3.2 Stochastic dynamics model

To model the dynamics in the video, we use a variational recurrent neural network (VRNN) [6]. The core of the dynamics model is a latent belief $\mathbf{z}$ over keypoint locations $\mathbf{x}$. In the VRNN architecture, the prior belief is conditioned on all previous timesteps through the hidden state $\mathbf{h}_{t-1}$ of an RNN, and thus represents a prediction of the current keypoint locations before observing the image:

$$p(\mathbf{z}_t | \mathbf{x}_{<t}, \mathbf{z}_{<t}) = \varphi^{\text{prior}}(\mathbf{h}_{t-1}) \tag{1}$$

We obtain the posterior belief by combining the previous hidden state with the unsupervised keypoint coordinates $\mathbf{x}_t = \varphi^{\text{det}}(\mathbf{v}_t)$ detected in the current frame:

$$q(\mathbf{z}_t | \mathbf{x}_{\leq t}, \mathbf{z}_{<t}) = \varphi^{\text{enc}}(\mathbf{h}_{t-1}, \mathbf{x}_t) \tag{2}$$

Predictions are made by decoding the latent belief:

$$p(\mathbf{x}_t | \mathbf{z}_{\leq t}, \mathbf{x}_{<t}) = \varphi^{\text{dec}}(\mathbf{z}_t, \mathbf{h}_{t-1}) \tag{3}$$

Finally, the RNN is updated to pass information forward in time:

$$\mathbf{h}_t = \varphi^{\text{RNN}}(\mathbf{x}_t, \mathbf{z}_t, \mathbf{h}_{t-1}). \tag{4}$$

Note that to compute the posterior (Eq. 2), we obtain $\mathbf{x}_t$ from the keypoint detector, but for the recurrence in Eq. 4, we obtain $\mathbf{x}_t$ by decoding the latent belief. We can therefore predict into the future without observing images by decoding $\mathbf{x}_t$ from the prior belief. Because the model has both deterministic and stochastic pathways across time, predictions can account for long-term dependencies as well as future uncertainty [10, 6].

## 4 Training

### 4.1 Keypoint detector

The keypoint detector is trained with a simple L2 image reconstruction loss $\mathcal{L}_{\text{image}} = \sum_t ||\mathbf{v} - \hat{\mathbf{v}}||_2^2$, where $\mathbf{v}$ is the true and $\hat{\mathbf{v}}$ is the reconstructed image. Errors from the dynamics model are not backpropagated into the keypoint detector.[2]

Ideally, the representation should use as few keypoints as possible to encode each object. To encourage such parsimony, we add two additional losses to the keypoint detector:

**Temporal separation loss.** Image features whose motion is highly correlated are likely to belong to the same object and should ideally be represented jointly by a single keypoint. We therefore add a separation loss that encourages keypoint trajectories to be decorrelated in time. The loss penalizes "overlap" between trajectories within a Gaussian radius $\sigma_{\text{sep}}$:

$$\mathcal{L}_{\text{sep}} = \sum_k \sum_{k'} \exp(-\frac{d_{kk'}}{2\sigma_{\text{sep}}^2}) \tag{5}$$

where $d_{kk'} = \frac{1}{T} \sum_t ||(\mathbf{x}_{t,k} - \langle \mathbf{x}_k \rangle) - (\mathbf{x}_{t,k'} - \langle \mathbf{x}_{k'} \rangle)||_2^2$ is the distance between the trajectories of keypoints $k$ and $k'$, computed after subtracting the temporal mean $\langle \mathbf{x} \rangle$ from each trajectory. $|| \cdot ||_2^2$ denotes the squared Euclidean norm.

**Keypoint sparsity loss.** For similar reasons, we add an L1 penalty $\mathcal{L}_{\text{sparse}} = \sum_k |\mu_k|$ on the keypoint scales $\mu$ to encourage keypoints to be sparsely active.

In Section 5.3, we show that both $\mathcal{L}_{\text{sep}}$ and $\mathcal{L}_{\text{sparse}}$ contribute to stable keypoint detection.

### 4.2 Dynamics model

The standard VRNN [6] is trained to encode the detected keypoints by maximizing the evidence lower bound (ELBO), which is composed of a reconstruction loss and a KL term between the Gaussian prior $\mathcal{N}_t^{\text{prior}} = \mathcal{N}(\mathbf{z}_t | \varphi^{\text{prior}}(\mathbf{h}_{t-1}))$ and posterior distribution $\mathcal{N}_t^{\text{enc}} = \mathcal{N}(\mathbf{z}_t | \varphi^{\text{enc}}(\mathbf{h}_{t-1}, \mathbf{x}_t))$:

$$\mathcal{L}_{\text{VRNN}} = -\sum_{t=1}^{T} \mathbb{E} \left[ \log p(\mathbf{x}_t | \mathbf{z}_{\leq t}, \mathbf{x}_{<t}) - \beta \text{KL}(\mathcal{N}_t^{\text{enc}} \| \mathcal{N}_t^{\text{prior}}) \right] \tag{6}$$

The KL term regularizes the latent representation. In the VRNN architecture, it is also responsible for training the RNN, since it encourages the prior to predict the posterior based on past information. To balance reliance on predictions with fidelity to observations, we add the hyperparameter $\beta$ (see also [2]). We found it essential to tune $\beta$ for each dataset to achieve a balance between reconstruction quality (lower $\beta$) and prediction diversity.

The KL term only trains the dynamics model for single-step predictions because the model receives observations after each step [10]. To encourage learning of long-term dependencies, we add a pure reconstruction loss, without the KL term, for multiple future timesteps:

$$\mathcal{L}_{\text{future}} = -\sum_{t=T+1}^{T+\Delta T} \mathbb{E} \left[ \log p(\mathbf{x}_t | \mathbf{z}_{\leq t}, \mathbf{x}_{\leq T}) \right] \tag{7}$$

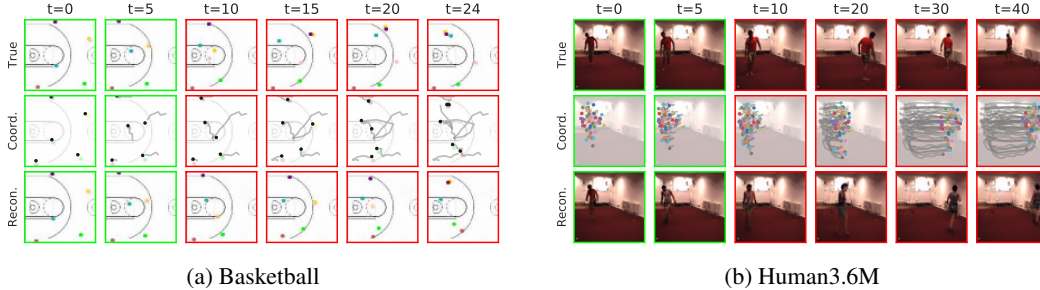

(a) Basketball
(b) Human3.6M

Figure 2: Main datasets used in our experiments. First row: Ground truth images. Second row: Decoded coordinates (black dots; $\hat{\mathbf{x}}_t$ in Figure 1) and past trajectories (gray lines). Third row: Reconstructed image. Green borders indicate observed frames, red indicate predicted frames.

The standard approach to estimate $\log p(\mathbf{x}_t | \mathbf{z}_{\leq t}, \mathbf{x}_{\leq t})$ in Eq. 6 and 7 is to sample a single $\mathbf{z}_t$. To further encourage diverse predictions, we instead use the best of a number of samples [4] at each timestep during training:

$$\max_i \big( \log p(\mathbf{x}_t | \mathbf{z}_{i,t}, \mathbf{z}_{<t}, \mathbf{x}_{<t}) \big), \tag{8}$$

where $\mathbf{z}_{i,t} \sim \mathcal{N}_t^{\mathrm{enc}}$ for observed steps and $\mathbf{z}_{i,t} \sim \mathcal{N}_t^{\mathrm{prior}}$ for predicted steps. By giving the model several chances to make a good prediction, it is encouraged to cover a range of likely data modes, rather than just the most likely. Sampling and evaluating several predictions at each timestep would be expensive in pixel space. However, since we learn the dynamics in the low-dimensional keypoint space, we can evaluate sampled predictions without reconstructing pixels. Due to the keypoint structure, the L2 distance of samples from the observed keypoints meaningfully captures sample quality. This would not be guaranteed for an unstructured latent representation. As shown in Section 5, the best-of-many objective is crucial to the performance of our model.

The combined loss of the whole model is:

$$\mathcal{L}_{\mathrm{image}} + \lambda_{\mathrm{sep}} \mathcal{L}_{\mathrm{sep}} + \lambda_{\mathrm{sparse}} \mathcal{L}_{\mathrm{sparse}} + \mathcal{L}_{\mathrm{VRNN}} + \mathcal{L}_{\mathrm{future}}, \tag{9}$$

where $\lambda_{\mathrm{sep}}$ and $\lambda_{\mathrm{sparse}}$ are scale parameters for the keypoint separation and sparsity losses. See Section S1 for implementation details, including a list of hyperparameters and tuning ranges (Table S1).

## 5  Results

We first show that the structured representation of our model improves prediction quality on two video datasets, and then show that it is more useful than unstructured representations for downstream tasks that require object-level information.

### 5.1  Structured representation improves video prediction

We evaluate frame prediction on two video datasets (Figure 2). The **Basketball** dataset consists of a synthetic top-down view of a basketball court containing five offensive players and the ball, all drawn as colored dots. The videos are generated from real basketball player trajectories [32], testing the ability of our model to detect and stably represent multiple objects with complex dynamics. The dataset contains 107,146 training and 13,845 test sequences. The **Human3.6** dataset [11] contains video sequences of human actors performing various actions. We use subjects S1, S5, S6, S7, and S9 for training (600 videos), and subjects S9 and S11 for evaluation (239 videos). For both datasets, ground truth object coordinates are available for evaluation, but are not used by the model. The Basketball dataset contains the coordinates of each of the 5 players and the ball. The Human dataset contains 32 motion capture points, of which we select 12 for evaluation.

We compare the full model (**Struct-VRNN**) to a series of baselines and ablations: the **Struct-VRNN (no BoM)** model was trained without the best-of-many objective; the **Struct-RNN** is deterministic; the **CNN-VRNN** architecture uses the same stochastic dynamics model as the Struct-VRNN, but uses an unstructured deep feature vector as its internal representation instead of structured keypoints. All structured models use $K = 12$ for Basketball, and $K = 48$ for Human3.6M, and were conditioned on

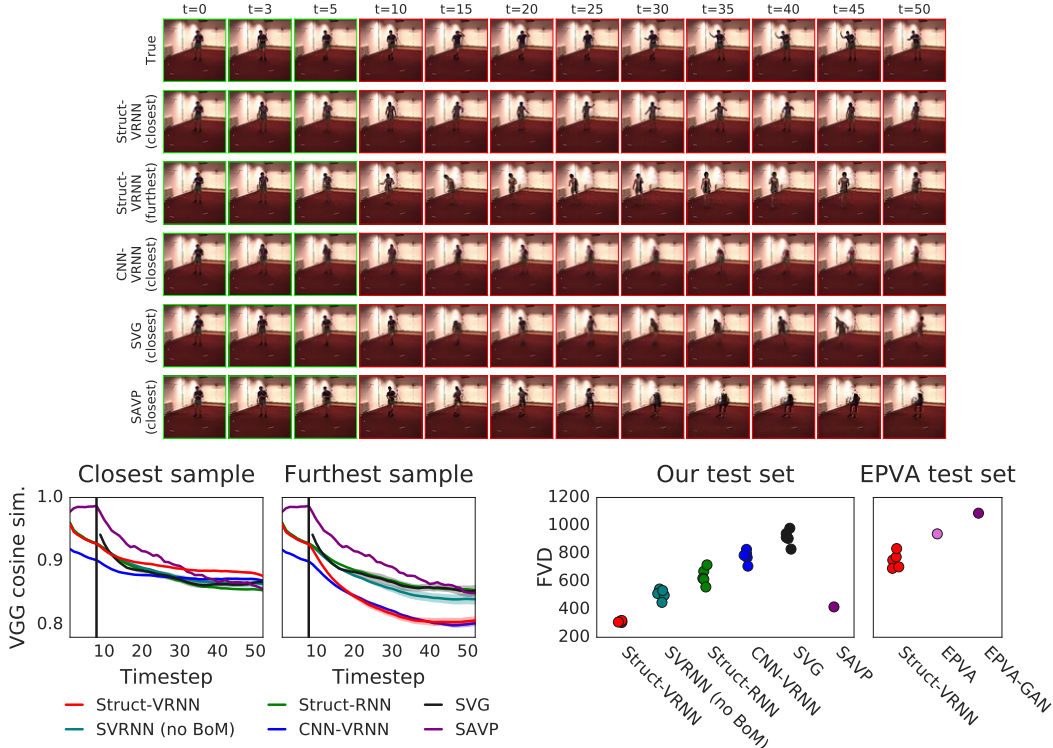

Figure 3: Video generation quality on Human3.6M. Our stochastic structured model (Struct-VRNN) outperforms our deterministic baseline (Struct-RNN), our unstructured baseline (CNN-VRNN), and the SVG [8] and SAVP [14] models. **Top:** Example observed (green borders) and predicted (red borders) frames (best viewed as video: `https://mjlm.github.io/video_structure/`). Example sequences are the closest or furthest samples from ground truth according to VGG cosine similarity, as indicated. Note that for Struct-VRNN, even the samples furthest from ground truth are of high visual quality. **Bottom left:** Mean VGG cosine similarity of the the samples closest to ground truth (left) and furthest from ground truth (right). Higher is better. Plots show mean performance across 5 model initializations, with the 95% confidence interval shaded. **Bottom right:** Fréchet Video Distance [23], using all samples. Lower is better. Dots represents separate model initializations. EPVA [28] is not stochastic, so we compare performance with a single sample from our method on their test set.

8 frames and trained to predict 8 future frames. For the CNN-VRNN, which lacks keypoint structure, we use a latent representation with $3K$ elements, such that its capacity is at least as large as that of the Struct-VRNN representation. Finally, we compare to three published models: **SVG** [8], **SAVP** [14] and **EPVA** [28] (Figure 3).

The Struct-VRNN model matches or outperforms the other models in perceptual image and video quality as measured by VGG [19] feature cosine similarity and Fréchet Video Distance [23] (Figure 3). Results for the lower-level metrics SSIM and PSNR are similar (see supplemental material).

The ablations suggest that the structured representation, the stochastic belief, and the best-of-many objective all contribute to model performance. The full Struct-VRNN model generates the best reconstructions of ground truth, and also generates the most diverse samples (i.e., samples that are furthest from ground truth; Figure 3 bottom left). In contrast, the ablated models and SVG show both lower best-case accuracy and smaller differences between closest and furthest samples, indicating less diverse samples. SAVP is closer, performing well on the single-frame metric (VGG cosine sim.), but still worse on FVD than the structured model. Qualitatively, Struct-VRNN exhibits sharper images and longer object permanence than the unstructured models (Figure 3, top; note limb detail and dynamics). This is true even for the samples that are far from ground truth (Figure 3 top, row "Struct-VRNN (furthest)"), which suggests that our model produces diverse high-quality samples, rather than just a few good samples among many diverse but unrealistic ones. This conclusion is

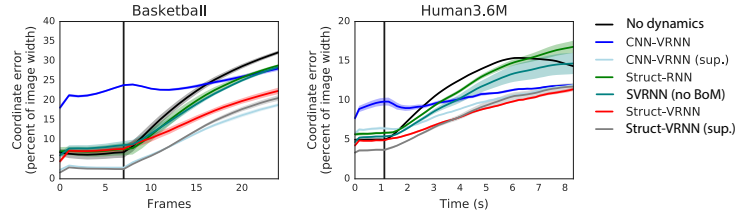

Figure 4: Prediction error for ground-truth trajectories by linear regression from predicted keypoints. (sup.) indicates supervised baseline.

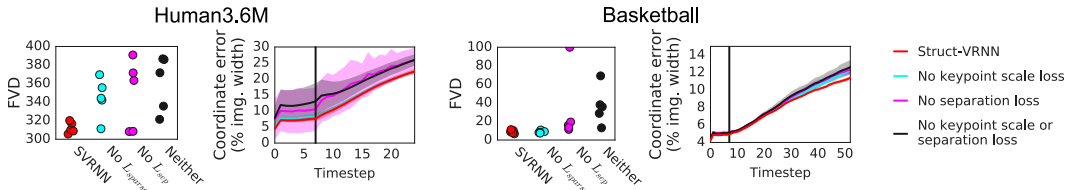

Figure 5: Ablating either the temporal separation loss or the keypoint sparsity loss reduces model performance and stability. In the FVD plots, each dot corresponds to a different model initialization. Coordinate error plots show the prediction error when regressing the ground-truth object coordinates on the discovered keypoints. Lines show the mean of five model initializations, with the 95% confidence intervals shaded.

backed up by the FVD (Figure 3 bottom right), which measures the overall quality of a distribution of videos [23].

## 5.2 The learned keypoints track objects

We now examine how well the learned keypoints track the location of objects. Since we do not expect the keypoints to align exactly with human-labeled objects, we fit a linear regression from the keypoints to the ground truth object positions and measure trajectory prediction error on held-out sequences (Figure 4). The trajectory error is the average distance between true and predicted coordinates at each timestep. To account for stochasticity, we sample 50 predictions and report the error of the best.[3]

As a baseline, we train Struct-VRNN and CNN-VRNN models with additional supervision that forces the learned keypoints to match the ground-truth keypoints. The keypoints learned by the unsupervised Struct-VRNN model are nearly as predictive as those trained with supervision, indicating that the learned keypoints represent useful spatial information. In contrast, prediction from the internal representation of the unsupervised CNN-VRNN is poor. When trained with supervision, however, the CNN-VRNN reaches similar performance as the supervised Struct-VRNN. In other words, both the Struct-VRNN and the CNN-VRNN can learn a spatial internal representation, but the Struct-VRNN learns it without supervision.

As expected, the less diverse predictions of the Struct-VRNN (no BoM) and Struct-RNN perform worse on the coordinate regression task. Finally, for comparison, we remove the dynamics model entirely and simply predict the last observed keypoint locations for all future timepoints. All models except unsupervised CNN-VRNN outperform this baseline.

## 5.3 Simple inductive biases improve object tracking

In Section 4.1, we described losses intended to add inductive biases such as keypoint sparsity and uncorrelated object trajectories to the keypoint detector. We find that these losses improve object tracking performance and stability. Figure 5 shows that models without $\mathcal{L}_{\text{sep}}$ and $\mathcal{L}_{\text{sparse}}$ show reduced video prediction and tracking performance. The increased variability between model initializations without $\mathcal{L}_{\text{sep}}$ and $\mathcal{L}_{\text{sparse}}$ suggests that these losses improve the learnability of a stable keypoint

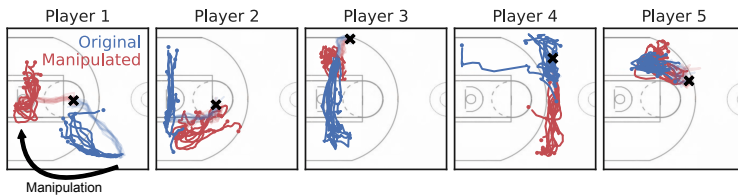

Figure 6: Unsupervised keypoints allow human-guided exploration of object dynamics. We manipulated the observed coordinates for Player 1 (black arrow) to change the original (blue) trajectory. The other players were not manipulated. The dynamics were then rolled out into the future to predict how the players will behave in the manipulated (red) scenario. Black crosses mark initial player positions. Light-colored parts of the trajectories are observed, dark-colored parts are predicted. Dots indicate final position. Lines of the same color indicate different samples conditioned on the same observed coordinates.

structure (also see Figure S6). In summary, we find that training and final performance is most stable if $K$ is chosen to be larger than the expected number of objects, such that the model can use $\mu$ in combination with $\mathcal{L}_{\text{sparse}}$ and $\mathcal{L}_{\text{sep}}$ to activate the optimal number of keypoints.

## 5.4 Manipulation of keypoints allows interaction with the model

Since the learned keypoints track objects, the model's predictions can be intuitively manipulated by directly adjusting the keypoints.

On the Basketball dataset, we can explore counterfactual scenarios such as predicting how the other players react if one player moves left as opposed to right (Figure 6). We simply manipulate the sequence observed keypoint locations before they are passed to the RNN, thus conditioning the RNN states and predictions on the manipulated observations.

For the Human3.6M dataset, we can independently manipulate body parts and generate poses that are not present in the training set (Figure 7; please see https://mjlm.github.io/video_structure/ for videos). The model

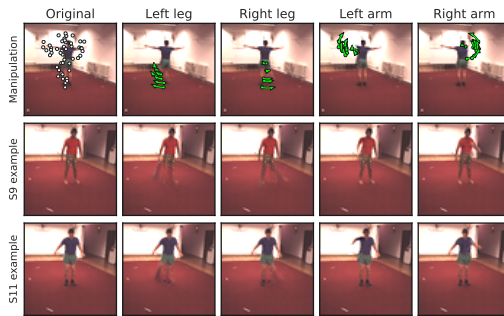

Figure 7: Keypoints learned by our method may be manipulated to change peoples' poses. Note that both manipulations and effects are spatially local. Best viewed in video (https://mjlm.github.io/video_structure/).

learns to associate keypoints with local areas of the body, such that moving keypoints near an arm moves the arm without changing the rest of the image.

## 5.5 Structured representation retains more semantic information

The learned keypoints are also useful for downstream tasks such as action recognition and reward prediction in reinforcement learning.

To test action recognition performance, we train a simple 3-layer RNN to classify Human3.6M actions from a sequence of keypoints (see Section S2.2 for model details).

The keypoints learned by the structured models perform better than the unstructured features learned by the CNN-VRNN (Figure 8). Future prediction is not needed, so the RNN and VRNN models perform similarly.

One major application we anticipate for our model is planning and reinforcement learning of spatially defined tasks. As a first step, we trained our model on a dataset collected

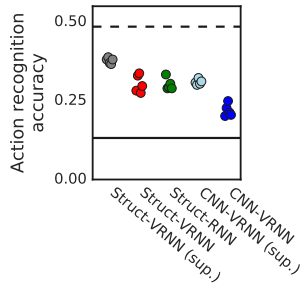

Figure 8: Action recognition on the Human3.6M dataset. Solid line: null model (predict the most frequent action). Dashed line: prediction from ground-truth coordinates. Sup., supervised.

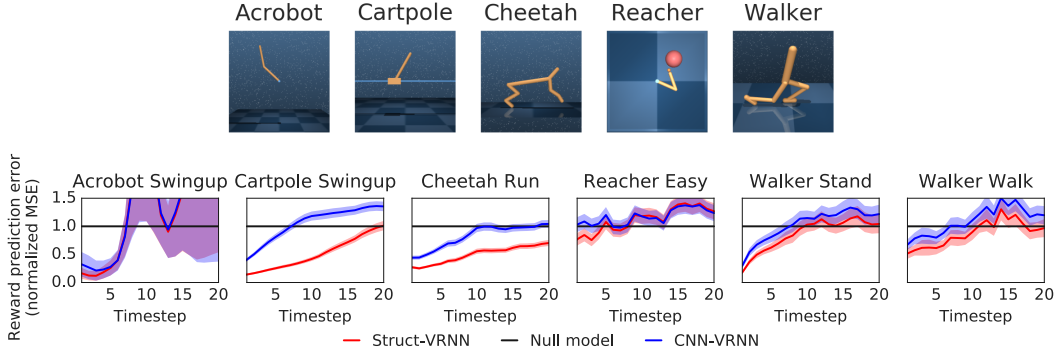

Figure 9: Predicting rewards on the DeepMind Control Suite continuous control domains. We chose domains with dense rewards to ensure the random policy would provide a sufficient reward signal for this analysis. To make scales comparable across domains, errors are normalized to a null model which predicts the mean training-set-reward at all timesteps. Lines show the mean across test-set examples and 5 random model initializations, with the 95% confidence interval shaded.

from six tasks in the DeepMind Control Suite (DMCS),
a set of simulated continuous control environments (Figure 9). Image observations and rewards were collected from the DMCS environments using random actions, and we modified our model to condition predictions on the agent's actions by feeding the actions as an additional input to the RNN. Models were trained without access to the task reward function. We used the latent state of the dynamics model as an input to a separate reward prediction model for each task (see Section S2.3 for details). The dynamics learned by the Struct-VRNN give better reward prediction performance than the unstructured CNN-VRNN baseline, suggesting our architecture may be a useful addition to planning and reinforcement learning models. Concurrent work that applies a similar keypoint-structured model to control tasks confirms these results [13].

# 6 Discussion

A major question in machine learning is to what degree prior knowledge should be built into a model, as opposed to learning it from the data. This question is especially important for unsupervised vision models trained on raw pixels, which are typically far removed from the information that is of interest for downstream tasks. We propose a model with a spatial inductive bias, resulting in a structured, keypoint-based internal representation. We show that this structure leads to superior results on downstream tasks compared to a representation derived from a CNN without a keypoint-based representational bottleneck.

The proposed spatial prior using keypoints represents a middle ground between unstructured representations and an explicitly object-centric approach. For example, we do not explicitly model object masks, occlusions, or depth. Our architecture either leaves these phenomena unmodeled, or learns them from the data. By choosing to not build this kind of structure into the architecture, we keep our model simple and achieve stable training (see variability across initializations in Figures 3, 4, and 5) on diverse datasets, including multiple objects and complex, articulated human shapes.

We also note the importance of stochasticity for the prediction of videos and object trajectories. In natural videos, any sequence of conditioning frames is consistent with an astronomical number of plausible future frames. We found that methods that increase sample diversity (e.g. the best-of-many objective [4]) led to large gains in FVD, which measures the similarity of real and predicted videos on the level of distributions over entire videos. Conversely, due to the diversity of plausible futures, frame-wise measures of similarity to ground truth (e.g. VGG cosine similarity, PSNR, and SSIM) are near-meaningless for measuring long-term video prediction quality.

Beyond image-based measures, the most meaningful evaluation of a predictive model is to apply it to downstream tasks of interest, such as planning and reinforcement learning for control tasks. Because of its simplicity, our architecture is straightforward to combine with existing architectures for tasks that may benefit from spatial structure. Applying our model to such tasks is an important future direction of this work.

## Footnotes

[2]We found this to be necessary to maintain a keypoint-structured representation. If the image model is trained based on errors from the dynamics model, the image model may adopt the poorly structured code of an incompletely trained dynamics model, rather than the dynamics model adopting the keypoint-structured code.

[3]For Human3.6M, we choose the best sample based on the average error of all coordinates. For Basketball, we choose the best sample separately for each player.

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
