[Supplementary Material]

# Unsupervised Learning of Object Structure and Dynamics from Videos
## (Supplemental material)

**Matthias Minderer**[*]   **Chen Sun**   **Ruben Villegas**   **Forrester Cole**
**Kevin Murphy**   **Honglak Lee**
Google Research
{mjlm, chensun, rubville, fcole, kpmurphy, honglak}@google.com

## S1   Model implementation details

### S1.1   Architecture

#### S1.1.1   Keypoint detector

Our goal is to encode the input image in terms of the locations of objects in the image [5, 9]. To this end, we use a **keypoint detector network** (Figure 1, bottom) that consists of $K$ keypoint detectors. Ideally, each detector learns to detect a distinct object or object part. The detector network is implemented as a series of convolutional layers with stride 2 that reduce the input image $\mathbf{v}$ into a stack of keypoint detection score maps with $K$ channels, $\mathbf{R} \in \mathbb{R}_{>0}^{H \times W \times K}$, where $H = W = 16$. We use the softplus function $f(x) = \log(1 + e^x)$ on the activations of the final layer to ensure the maps are positive.

The raw maps $\mathbf{R}$ are normalized to obtain detection weight maps $\mathbf{D}$,

$$\mathbf{D}_k(u, v) = \frac{\mathbf{R}_k(u, v)}{\sum_u \sum_v \mathbf{R}_k(u, v)}, \tag{S1}$$

where $\mathbf{D}_k(u, v)$ is the value of the $k$-th channel of $\mathbf{D}$ at pixel $(u, v)$. We then reduce each $\mathbf{D}_k$ to a single $(x, y)$-coordinate by computing the weighted mean over pixel coordinates:

$$(x_k, y_k) = \sum_{v=1}^{H} \sum_{u=1}^{W} (u, v) \cdot \mathbf{D}_k(u, v) \tag{S2}$$

To model keypoint presence or absence, we compute the mean value of the raw detection score maps,

$$\mu_k = \frac{1}{H \times W} \sum_{v=1}^{H} \sum_{u=1}^{W} \mathbf{R}_k(u, v). \tag{S3}$$

In summary, each keypoint is represented by a $(x, y, \mu)$-triplet encoding its location in the image and its scale.

For **image reconstruction**, each keypoint is converted back into a pixel representation by creating a map $\hat{\mathbf{R}}_k$ containing a Gaussian blob with standard deviation $\sigma_{\text{k.p.}}$ at the location of the keypoint, scaled by $\mu_k$:

$$\hat{\mathbf{R}}_k(u, v) = \mu_k \cdot \exp\left(-\frac{1}{2\sigma_{\text{k.p.}}^2} ||(u, v) - (x_k, y_k)||^2\right). \tag{S4}$$

---

[*]Google AI Resident

The map $\hat{\mathbf{R}}_k$ contains the same information as the keypoint tuple $(x_k, y_k, \mu_k)$, but in a pixel representation that is suitable as input to the convolutional reconstructor network $\varphi^{\text{rec}}$. The image is reconstructed as follows:

$$\hat{\mathbf{v}}_t = \mathbf{v}_1 + \varphi^{\text{rec}}([\hat{\mathbf{R}}_t, \hat{\mathbf{R}}_1, \varphi^{\text{appearance}}(\mathbf{v}_1)]) \tag{S5}$$

where $\varphi^{\text{rec}}$ applies alternating convolutional layers and twofold bilinear upsampling until the $16 \times 16$ maps are expanded to the original image resolution, $[\cdots]$ denotes concatenation (here, channel-wise), and $\varphi^{\text{appearance}}$ is a network with the same architecture as $\varphi^{\text{det}}$ (except for the final softmax nonlinearity) that extracts image features from the first frame $\mathbf{v}_1$ to capture appearance information of the scene.

The internal layers of the convolutional encoder and decoder are connected through leaky rectified linear units $f(x) = \max(x, 0.2x)$. L2 weight decay of $10^{-4}$ is applied to all convolutional kernels. To increase model capacity, we add one (for Basketball and DMCS) or two (for Human3.6M) additional size-preserving (stride 1) convolutional layers at each resolution scale of the detector and reconstructor. The image resolution is $64 \times 64$ pixels.

### S1.1.2 Dynamics model

The dynamics model (Figure 1, top) has the following components:

The **prior** network consists of a dense layer with ReLU activation functions (for number of units, see *Prior net size* in Table S1), followed by a dense layer that projects the activations to the mean and standard deviation that parameterize the prior latent distribution $\mathcal{N}_t^{\text{prior}}$,

$$\boldsymbol{\mu}_t^{\text{prior}}, \boldsymbol{\sigma}_t^{\text{prior}} = \varphi^{\text{prior}}(\mathbf{h}_{t-1}). \tag{S6}$$

The **encoder** network consists of a dense layer with 128 units and ReLU activation functions, followed by a dense layer that projects the activations to the mean and standard deviation that parameterize the posterior latent distribution $\mathcal{N}_t^{\text{enc}}$,

$$\boldsymbol{\mu}_t^{\text{enc}}, \boldsymbol{\sigma}_t^{\text{enc}} = \varphi^{\text{enc}}([\mathbf{x}_t, \mathbf{h}_{t-1}]). \tag{S7}$$

The **decoder** network consists of a dense layer with 128 units and ReLU activation functions, followed by a dense layer that projects the activations to the the linearized keypoint vector $\mathbf{x}_t$ of length $K \times 3$ (containing $x$, $y$ and $\mu$ components),

$$\mathbf{x}_t = \varphi^{\text{dec}}([\mathbf{z}_t, \mathbf{h}_{t-1}]), \tag{S8}$$

where $\mathbf{z}_t \sim \mathcal{N}^{\text{enc}}(\boldsymbol{\mu}_t, \boldsymbol{\sigma}_t^2 \mathbb{I})$ for observed steps and $\mathbf{z}_t \sim \mathcal{N}^{\text{prior}}(\boldsymbol{\mu}_t, \boldsymbol{\sigma}_t^2 \mathbb{I})$ for predicted steps.

The **recurrent** network consists of a GRU layer with 512 units:

$$\mathbf{h}_t = \varphi^{\text{rnn}}([\mathbf{x}_t, \mathbf{z}_t, \mathbf{h}_{t-1}]). \tag{S9}$$

For the action-conditional model used for reward prediction (Figure 9), the input to $\varphi^{\text{rnn}}$ is $[\mathbf{x}_t, \mathbf{z}_t, \mathbf{h}_{t-1}, \mathbf{a}_{t-1}]$, where $\mathbf{a}_{t-1}$ is the vector of random actions used to generate frame $t$ of the DeepMind Control Suite dataset.

The size of $\varphi^{\text{prior}}$ and the latent representation $\mathbf{z}$ were optimized as hyperparameters (see Table S1).

### S1.2 Optimization

We used the ADAM optimizer [6] with $\beta_1 = 0.9$ and $\beta_2 = 0.999$. We trained on batches of size 32 for $10^5$ steps. The learning rate was set to $10^{-3}$ at the start of training and reduced by half every $3 \times 10^4$ steps. We used an L2 weight decay of $10^{-4}$ on the weights of the convolutional layers in the image encoder and decoder. Weights were initialized using the "He uniform" method as implemented by Keras. Models were trained on a single Nvidia P100 GPU. Training took approximately 12 hours.

During training, we linearly annealed the KL loss scale from 0 to $\beta$ over the first $2.5 \times 10^4$ steps, as in [2].

Table S1: Hyperparameters

| Parameter name | Symbol | Tuning range | Basketball | Human3.6M | DMCS |
|---|---|---|---|---|---|
| Batch size | | - | 32 | 32 | 32 |
| Init. learning rate | | - | $10^{-3}$ | $10^{-3}$ | $10^{-3}$ |
| Input steps | $T$ | - | 8 | 8 | 8 |
| Predicted steps | $\Delta T$ | $[0, 32]$ | 8 | 8 | 8 |
| Num. keypoints | $K$ | varied | 12 | 48 | 64 |
| Keypoint sparsity scale | $\lambda_{\text{sparse}}$ | $[10^{-3}, 10^4]$ | 0.1 | $10^{-2}$ | 5 |
| Separation loss scale | $\lambda_{\text{sep}}$ | $[10^{-3}, 10^4]$ | 0.1 | $2 \times 10^{-2}$ | 0.1 |
| Separation loss width | $\sigma_{\text{sep}}$ | $[0, 0.2]$ | $2 \times 10^{-2}$ | $2 \times 10^{-3}$ | $2 \times 10^{-2}$ |
| Keypoint blob width | $\sigma_{\text{k.p.}}$ | $[0.1, 2.0]$ (pix) | 1.5 | 1.5 | 1.5 |
| Latent code size | | $[4, 256]$ | 16 | 16 | 128 |
| KL loss scale | $\beta$ | $[10^{-3}, 10]$ | $10^{-2}$ | $10^{-2}$ | $3 \times 10^{-3}$ |
| Prior net size | | $[4, 512]$ | 16 | 4 | 16 |
| Posterior net size | | - | 128 | 128 | 128 |
| Num. RNN units | | $[32, 2048]$ | 512 | 512 | 512 |
| Num. samp. for BoM loss | | $[1, 200]$ | 50 | 50 | 50 |

## S1.3    Scheduled sampling

When training an RNN for many timesteps, the initially large errors compound over time, leading to slow learning. Therefore, during training, we initially supplied the observed keypoint coordinates as $\mathbf{x}_{t-1}$ to the RNN, instead of the RNN's own predictions. This is similar to teacher forcing, although we note that we used the output of the unsupervised keypoint detector, rather than the ground truth.

We find that teacher forcing causes the model to make more dynamic predictions which are qualitatively realistic, but may have poor error metrics because of the mismatch between the training and test distributions. We therefore gradually switched to using samples from the model over the course of training (scheduled sampling, [1]). We linearly increased the probability of choosing samples from the model from 0 to a final value over the course of training. We chose the final probability to be 1.0 for the observed timesteps and 0.5 for the predicted timesteps.

## S1.4    Hyperparameter optimization

We used a black-box optimization tool based on Gaussian process bandits [4] to tune several of the hyperparameters of our model. The target for optimization was the mean coordinate trajectory error as computed for Figure 4. See Table S1 for parameters and their tuning ranges.

# S2    Experimental details

## S2.1    Comparison to SVG and SAVP

For both SVG and SAVP, models were trained on the same datasets as our models, using the code made available by the authors. We trained all models using 8 input steps and 8 predicted steps. For SAVP, the other hyperparameters were set to those recommended by the authors for the BAIR robot pushing dataset.

## S2.2    Human3.6M action recognition

To understand how much semantically useful information the representations of our models contain, we predicted the actions performed in the Human3.6M dataset from the model representations (Figure 8). We first used trained models to extract keypoints (or unstructured image representations) for sequences of 8 observed steps from the Human3.6M test set. These keypoint sequences represented the dataset used for action recognition. The action recognition training set comprised 881 sequences, the test set 279 sequences. We ensured that no test sequences came from the same original videos as those used in the training set.

We then trained a separate recurrent neural network to classify each sequence into one of the 15 action categories (Walking, Sitting, Eating, Discussions, ...) in the Human3.6M dataset. No categories were excluded. The network consisted of two GRU layers (128 units), followed by a dense layer (15 units) and a softmax layer. We used 25% dropout after each GRU layer. The model was trained for 100 epochs using the ADAM optimizer with a starting learning rate of 0.01 that was successively reduced to 0.0001. We report the mean action recognition accuracy (fraction correct) on the 279 test sequences.

### S2.3  DeepMind Control Suite reward prediction

To explore if the structured representation learned by our model may be useful for planning, we used our model to predict rewards in DeepMind Control Suite [7] continuous control tasks (Figure 9). We chose tasks that have dense rewards and thus provide a strong signal for evaluation (Acrobot Swingup, Cartpole Balance, Cheetah Run, Reacher Easy, Walker Stand, Walker Walk).

We generated a dataset based on DeepMind Control Suite (DMCS) continuous control tasks by performing random actions and recording 64 by 64 pixel observations, the actions, and the rewards. We then trained our model variants on this dataset. Importantly, we trained a single model on data from all domains, to test the generality of our approach. We modified our models to be action-conditional by passing the vector of actions as an additional input to the RNN at each timestep.

To predict rewards, we used the RNN hidden state of our models as a representation of the dynamics learned by the model. We first collected the hidden states of the trained models for 10,000 length-20 sequences from the test split for each of the six domains in our DMCS dataset. We then trained a separate, smaller reward prediction model to predict rewards for each of the six domains. The reward prediction models took the sequence of RNN hidden states as input and returned a sequence of scalar reward values as output. The model consisted of a fully connected layer (128 units), two GRU layers (128 units) and a dense layer (1 unit), all connected through rectified linear units. The reward prediction model was trained on 80% of the data with the ADAM optimizer with a starting learning rate of 0.001 that was successively reduced to 0.0001. We report the mean squared error of the predicted reward on the remaining 20% of the data.

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

Figure S1: Additional video metrics on Human3.6M: structural similarity (SSIM) and peak signal-to-noise ratio (PSNR). The models were conditioned on 8 frames and trained to predict 8 future frames. Higher is better (closer to ground truth). Top row shows the mean across all test-set examples of the closest-to-GT of 100 stochastic samples, bottom shows the furthest. Lines show the mean across 5 random model initializations, with the 95% confidence interval shaded.

Figure S2: Video generation quality on Basketball. The models were conditioned on 8 frames and trained to predict 8 future frames. Our stochastic structured model (Struct-VRNN) outperforms our deterministic baseline (Struct-VRNN), our unstructured baseline (CNN-VRNN), and the SVG model [3] in the FVD metric and qualitatively. **Top:** Example input (green borders) and predicted (red borders) frames. **Bottom left:** Fréchet Video Distance (FVD) [8]. Lower is better. Each dot represents a separate model initialization. For SVG, the FVD for several runs was greater than 1700. The example at the top comes from the best run. **Bottom right:** VGG feature cosine similarity, structural similarity (SSIM), and peak signal-to-noise ratio (PSNR). Higher is better. Lines show the mean across 5 random model initializations, with the 95% confidence interval shaded. The SVG model fails to represent objects stably at later timepoints. This is captured by the FVD metric, causing a large difference to our models. However, it is not captured by the other metrics, suggesting that they are not informative on this synthetic dataset. Also see videos in supplemental material or at `https://mjlm.github.io/video_structure/`.

(a) Struct-VRNN

(b) Struct-VRNN without best-of-many-samples objective

(c) Struct-RNN (deterministic dynamics)

Figure S3: Effect of stochastic belief and best-of-many-samples objective on sample diversity. Each row shows one example Basketball play, with the trajectories for one player in each column. The black line indicates the true trajectory, the colored lines indicate 20 stochastic predictions, all conditioned on the same observed steps. Trajectory endpoints are marked with dots. The model trained with the best-of-many-samples objective (a) produces more diverse samples than the model without (b). As expected, the deterministic model (c) lacks diversity completely. Players were matched to detected keypoints by finding, for each player, the keypoint which was closest to that player on average.

Figure S4: The Struct-VRNN model generates plausible and diverse predictions. Each block shows the true sequence in the top row, followed by three samples conditioned on the same initial frames (green outlines). Also see videos in supplemental material or at `https://mjlm.github.io/video_structure/`.

(a) Acrobot

(b) Cartpole

(c) Cheetah

(d) Reacher

(e) Walker

Figure S5: Action-conditional predictions for the DeepMind Control Suite domains. Even though the CNN-VRNN has enough capacity to encode the observed frames (green outlines) well, it struggles to make future predictions (red outlines), in contrast to the Struct-VRNN. Also see videos in supplemental material or at `https://mjlm.github.io/video_structure/`.

(a) Coordinate error over time for individual objects. In the left plot, lines indicate the mean across 10 model initializations, with the 95% confidence interval shaded. In the middle and right plot, lines show individual model initializations. For these runs, $\lambda_{\text{sparse}} = 0.01$ and $\lambda_{\text{sep}} = 0.01$.

(b) Mean coordinate error of the ball (left) and Player 3 (right) across different settings of $\lambda_{\text{sparse}}$ and $\lambda_{\text{sep}}$, relative to the best settings. Each entry in the heatmaps corresponds to the mean trajectory error across time and 10 model initializations.

Figure S6: Analysis of object tracking failure modes (Basketball dataset). (a) shows the coordinate error for individual objects. We identify two different failure modes, corresponding to failures of the dynamics model and failures of the keypoint detector: Some objects, e.g. the ball (yellow; middle plot), have relatively large tracking errors across all 10 model initializations, presumably because their dynamics are hard to learn. Other objects, e.g. Player 3 (pink; right plot) are tracked well in some and poorly in other model initializations, presumably because the keypoint detector completely fails to detect these objects in some runs. The sweep over $\lambda_{\text{sparse}}$ and $\lambda_{\text{sep}}$ in (b) shows that $\mathcal{L}_{\text{sparse}}$ and $\mathcal{L}_{\text{sep}}$ primarily reduce the keypoint detector failure mode (exemplified by the Player 3 error; right), while the tracking error of the ball (left) is insensitive to these losses. In other words, $\mathcal{L}_{\text{sparse}}$ and $\mathcal{L}_{\text{sep}}$ improve the reliability of the keypoint detector.