[Reviews · NeurIPS 2019]

Reviewer 1



To my knowledge, the concept of performing video frame prediction from a sparse keypoint representation is novel. However, in my opinion, the technical novelty in the paper is somewhat thin. The unsupervised keypoint detector, proposed in [12], is being reused with little modifications. Same thing can be said about the VRNN model. The differences that I can see are the temporal separation loss and keypoint sparsity loss in keypoint detector training. However, without enforcing temporal consistency and matching (i.e. optical flow or tracking), the keypoints can "jump" around between frames, how do you deal with this problem? Second, choosing a large k (sparsity loss) at the beginning will interfere with the temporal loss, did you observe any issues in the training process? Finally, what is the size of the feature vector in CNN-VRNN? My assumption is that the size of the feature vector in CNN-VRNN would be smaller than a K x 3 vector of keypoints, thus the performance gains could come from a increase in information being stored in the feature vector. Other than that, I think the authors did a great job to evaluate their method through detailed experiments. I have no comments about the quality of writing and presentation. Although I would prefer a stronger line stroke for the proposed method to emphasize the results in the plots (Figure 3, 4, and 7), and use box plots for Figure 3 and 8. Figure 3 (bottom left) has a small text (XID: 5985036) in the plot.

Reviewer 2



In this paper the authors propose a novel deep architecture for predicting future video frames. This builds on a variational recurrent neural network and combines several novel features including a keypoint representation and a best of many prediction scheme. The authors validate this on several datasets showing improvements over competing approaches. Overall this is a reasonably good paper - most of my comments are relatively minor. - It is hard to see how the keypoint construction approach actually imposes spatial structure. My feeling is that this should be very sensitive to initialization, but this seems not to be the case. Can the authors explain how this can be so? - Some papers (e.g. ref [27]) suggests that adversarial training may improve prediction. Why do the authors not compare to that? - Line 124: can the authors provide more intuition about why this should be the case? did the authors also try with the linkkng the two modules?

Reviewer 3



Originality: The main contribution of the paper is to propose a structured representation for video prediction models based on extracting keypoints from images. Models that extract keypoints from images had been proposed before, and here the authors propose an extension of those ideas to video. The paper also has experiments to empirically analyze this representation, which is often lacking in other video prediction papers, despite the fact that learning representations is one of the main motivations for video prediction. Clarity: The paper is well organized and clearly written. Quality and significance: The experiments are sound and properly assess some of the points made by the authors. I believe there are some issues/typos with the model formulation. The likelihood term in equation 2 should not include x_t in the condition (otherwise p(x_t|x_t, other RVs) is trivial. Similarly for equation 4. I also think that the claims that this model avoids compounding errors and that it has efficient sampling are a bit misleading/not properly supported. As they are described at the moment they are not particular to using a keypoint representation. Instead, they are caused by the decoupled training of the encoder-decoder and the VRNN dynamics model, but this could be done also with unstructured frame representations. I also think that there should be an ablation of the different extra losses in the main paper. The experiments show an improvement using the 'best of many' technique. In the supplementary material figure S1 shows that there's a slight improvement for downstream tasks using the different losses. However, this should be quantified for video prediction too. Also note that some of the extra losses (sparsity of the representation) could be adapted to unstructured representations, and I wonder how would this change the results for downstream tasks. As for the experiments and their results, I think some conclusions extracted by the authors are a bit unjustified. For video prediction (figure 3), in FVD the model has a clear advantage, but not so much in terms of VGG cosine similarity, where it seems that the model's performance has a lot of variance (best in terms of closest sample but worse than baselines in terms of the furthest sample). Note that at sampling time without ground truth having such high variance could mean that some samples are very poor. While the authors argue that this is a sign of increased diversity, this is not trivial nor properly supported. In practice it means that some of the samples generated by the model deviate quite a lot from the ground truth. Furthermore the authors do not compare to other contemporary state of the art methods such as SAVP [13], which obtains significantly better FVD scores than the SVG baseline. It would be interesting to include such comparison. In general, despite the above issues, I believe they do not significantly alter the conclusions from the experiments and therefore I am in favor of accepting the paper for its novelty and positive results. Minor notes: Figure 7 graph 1 has wrong x/y limits, the lines go out of the plot. line 126 typo: 'ideally, the representation should (missing verb) as few keypoints...' ------------- POST REBUTTAL UPDATE ---------------------- I read the rebuttal and the other reviews. Some of my comments such as the one regarding compounding errors have not been addressed, however I still believe the paper is a good contribution and keep my rating.

Reviewer 4



Originality: * The approach seems novel. * Related work seems adequately cited. Quality: * The method seems technically sound though the architecture details should be presented in the main paper. Some issues: - l. 223, it is not clear if the RNN cells have memory/hidden states and this affects the ability to alter the prediction by changing the keypoints. Please add an explanation why it can work. - Eq 1, rhs should be negated for a loss. - l. 191, the larger diversity of quality of the samples is a shortcoming of the proposed method. Apparent from Fig. 3 left is that the proposed method can perform significantly worse for some samples than other sota methods. A plot also showing mean/std dev for the variants might be insightful, please add. How to choose good samples at test time? Please discuss. * Sec 5.2, a more finegrained evaluation would be interesting here, if the learned keypoints can capture all objects in the scene, or if it chooses to represent only a few and which. Does the prediction error contain outliers for some objects? * Please define or revise the term "object structure" in the title. The title is too generic, please mention keypoints in the title. Clarity: * The paper is very well written and easy to follow. Technical details about network architecture should be included in the main paper. Significance: * Latent dynamics representations of video are highly relevant for video prediction and planning. The proposed approach is interesting, since it suggests a way to add structure or inductive bias by predefining that the learning algorithm needs to use a number of localized keypoints for describing the video content.

[Author Response · NeurIPS 2019]

We thank the reviewers for their thoughtful comments and suggestions. We performed several new experiments and analyses to address the comments and will make the suggested changes to the main text. We also thank all reviewers for taking the time to point out minor errors. Below, we address the reviewers' comments individually.

**R1, R6: Additional analyses/ablations for $\mathcal{L}_{\mathbf{sparse}}$ and $\mathcal{L}_{\mathbf{sep}}$.** We agree with Reviewer 1 that much of the novelty of our work lies in the losses and training approach. We performed new analyses to show that $\mathcal{L}_{\text{sparse}}$ and $\mathcal{L}_{\text{sep}}$ are crucial to the performance and stability of the model, both in terms of video metrics (especially FVD, Fig. A) and coordinate tracking accuracy (Fig. B), on which downstream tasks depend. We will add these analyses to the main text.

**R1: Temporal consistency and "jumping" keypoints.** We initially experimented with using predictions from the dynamics model as "prior" for the keypoint detector, but achieved better performance without enforcing temporal consistency explicitly. Keypoints can indeed "jump" between frames, but we show in a new analysis (Fig. D) that the VRNN partially smooths over such jumps: We displaced the location of one keypoint by $0.5 \times$ image width in the direction of the image center for one frame (Basketball dataset). The keypoint location inferred by the VRNN jumps by less than $0.5 \times$ image width in the perturbed frame and quickly recovers. Jumping thus seems to be a minor issue.

**R1: Did you observe training issues when combining a large $K$ with $\mathcal{L}_{\mathbf{sep}}$?** Note that the optimal $\sigma_{\text{sep}}$ (spatial Gaussian radius of $\mathcal{L}_{\text{sep}}$) is very small ($\sigma_{\text{sep}} = 2 \times 10^{-3} \times$ image width for Human3.6M). At this $\sigma_{\text{sep}}$, the loss does not interfere with initial training even for large $K$, but still prevents keypoints from collapsing onto the same image feature.

**R1: What is the size of the feature vector in CNN-VRNN?** We made sure to match the size of the feature vectors of the models, such that the CNN-VRNN had $K \times 3$ dimension at the narrowest point. Therefore, in principle, the CNN-VRNN had the capacity to exactly recapitulate the Struct-VRNN structure.

**R1: Usefulness of KP structure for RL.** Our claim has since been confirmed by Kulkarni et al. (arXiv 1906.11883v1).

**R5: How is spatial structure imposed and why is it not sensitive to initialization?** See Jakab et al. [12] for how the keypoint detector imposes spatial structure. A naïve application of [12] to video indeed suffers from sensitivity to initialization (see Figs. A and B, "no $\mathcal{L}_{\text{sparse}}/\mathcal{L}_{\text{sep}}$ loss"). By adding $\mathcal{L}_{\text{sparse}}$ and $\mathcal{L}_{\text{sep}}$, we achieve high robustness.

**R5, R6: Comparison to adversarial methods.** We note that we do compare to an adversarial method ("EPVA-GAN", Fig. 3, bottom right). A GAN loss could also be added to our model as a complementary objective; this is orthogonal to our contributions. We agree that comparison to SAVP would be interesting, but we could not obtain results in time for the rebuttal. We will include them in the final paper.

**R5: Why train keypoint detector and dynamics model separately?** We initially tried to train the model jointly ($\varphi^{\text{det}} \rightarrow$ VRNN $\rightarrow \varphi^{\text{rec}}$), but found that the model learned an unstructured latent code, rather than spatially meaningful keypoints. Presumably it was easier for $\varphi^{\text{rec}}$ to reconstruct the image from an unstructured code, than for the VRNN to learn the keypoint structure. Isolating the keypoint detector from the dynamics model solves this problem.

**R6: Why not apply B.o.M. sampling and $\mathcal{L}_{\mathbf{sparse}}$ to CNN-VRNN?** We did apply both to CNN-VRNN, but this yields no gains because sample evaluation and sparsity are less meaningful in an unstructured space than in keypoint space.

**R6, R7: Is sample diversity an advantage? Are all samples good?** We agree with Reviewers 6 and 7 that we need to expand the discussion of sample diversity. Fig. E below shows that even the samples with the lowest VGG cosine similarity to ground truth are of high visual quality. For videos, see Sections 2 and 3 on the anonymous website (link in original submission). We will add more examples and videos to the final paper. We emphasize that frame-wise similarity to GT (e.g. VGG sim, PSNR and SSIM) does not meaningfully measure video prediction quality. For real data, at test time, there is no single "ground truth". Instead, there is an astronomical number of plausible futures that are all consistent with the conditioning frames. We believe that most previous models dramatically underestimate this diversity; our model comes closer to it. This is backed up by FVD, which is designed to measure sample diversity.

**R7: More fine-grained evaluation of object tracking.** We performed a new analysis of per-object tracking performance on Basketball (Fig. C). We identified two different failure modes: The basketball (yellow traces) has relatively large tracking errors across all 10 model initializations, presumably because the dynamics of the ball are hard to learn. On the other hand, Player 3 (pink) is tracked well in some and poorly in other model initializations, presumably because the keypoint detector fails to recognize the light-colored object. We will describe these failure modes in the main text.

Figure A: Video metrics for loss ablations. Each dot is one model initialization.

Figure B: Coordinate regression error for 10 model initializations (Basketball dataset).

Figure C: Different objects (e.g. Ball and Player 3 in Basketball) have different error modes.

Figure D: The dynamics model partially smooths over "jumping" keypoints. We displaced the location of one object by $0.5 \times$ img. width. The location inferred by the VRNN jumps by less than $0.5 \times$ img. width in the perturbed frame and quickly recovers (Basketball dset).

Figure E: Even the samples with the lowest VGG cosine similarity to ground truth are visually realistic.

[Meta-Review · NeurIPS 2019]

The paper proposes a new model for video prediction with a structured representation based on object keypoints. It is a novel approach and also experiment methodology is interesting and generalizable. Reviewers initially asked many questions and the rebuttal was convincing, at least for the majority of reviewers. Thus according with their discussion the area chair suggest an acceptance